# Identification of Transcription Factors Responsible for a Transforming Growth Factor-β-Driven Hypertrophy-like Phenotype in Human Osteoarthritic Chondrocytes

**DOI:** 10.3390/cells11071232

**Published:** 2022-04-05

**Authors:** Nathalie G. M. Thielen, Margot Neefjes, Elly L. Vitters, Henk M. van Beuningen, Arjen B. Blom, Marije I. Koenders, Peter L. E. M. van Lent, Fons A. J. van de Loo, Esmeralda N. Blaney Davidson, Arjan P. M. van Caam, Peter M. van der Kraan

**Affiliations:** Department of Experimental Rheumatology, Radboud Institute for Molecular Life Sciences, Radboud University Medical Center, 6525 GA Nijmegen, The Netherlands; nathalie.thielen@radboudumc.nl (N.G.M.T.); margot.neefjes@radboudumc.nl (M.N.); elly.vitters@radboudumc.nl (E.L.V.); henk.vanbeuningen@radboudumc.nl (H.M.v.B.); arjen.blom@radboudumc.nl (A.B.B.); marije.koenders@radboudumc.nl (M.I.K.); peter.vanlent@radboudumc.nl (P.L.E.M.v.L.); fons.vandeloo@radboudumc.nl (F.A.J.v.d.L.); esmeralda.blaneydavidson@radboudumc.nl (E.N.B.D.); arjan.vancaam@radboudumc.nl (A.P.M.v.C.)

**Keywords:** chondrocyte hypertrophy, TGF-β, osteoarthritis, transcription factors, ALK5

## Abstract

During osteoarthritis (OA), hypertrophy-like chondrocytes contribute to the disease process. TGF-β’s signaling pathways can contribute to a hypertrophy(-like) phenotype in chondrocytes, especially at high doses of TGF-β. In this study, we examine which transcription factors (TFs) are activated and involved in TGF-β-dependent induction of a hypertrophy-like phenotype in human OA chondrocytes. We found that TGF-β, at levels found in synovial fluid in OA patients, induces hypertrophic differentiation, as characterized by increased expression of *RUNX2*, *COL10A1*, *COL1A1*, *VEGFA* and *IHH*. Using luciferase-based TF activity assays, we observed that the expression of these hypertrophy genes positively correlated to SMAD3:4, STAT3 and AP1 activity. Blocking these TFs using specific inhibitors for ALK-5-induced SMAD signaling (5 µM SB-505124), JAK-STAT signaling (1 µM Tofacitinib) and JNK signaling (10 µM SP-600125) led to the striking observation that only SB-505124 repressed the expression of hypertrophy factors in TGF-β-stimulated chondrocytes. Therefore, we conclude that ALK5 kinase activity is essential for TGF-β-induced expression of crucial hypertrophy factors in chondrocytes.

## 1. Introduction

Osteoarthritis (OA) is the result of a disturbance in joint homeostasis, leading to a loss of articular cartilage. In articular chondrocytes, this disturbance is associated with a switch towards a hypertrophy-like phenotype, which resembles the terminal differentiation route in growth plate chondrocytes [1,2,3]. Such hypertrophy-like cells express factors such as collagen type 10 (*COL10A1*), runt-related transcription factor 2 (*RUNX2*), Indian hedgehog (*IHH*) and vascular endothelial growth factor A (*VEGFA*) [2,4,5]. The presence of hypertrophic chondrocytes is thought to contribute to the disease process by production of cartilage-degrading enzymes such as metalloproteinases (MMPs) and aggrecanases (ADAMTS) [4,5].

In healthy cartilage, transforming growth factor-β (TGF-β) prevents chondrocyte hypertrophic differentiation and maintains chondrocyte homeostasis [6,7,8]. It does this via activation of the intracellular transcription factors (TFs) SMAD2 and SMAD3 [9,10]. At the same time, the literature indicates that TGF-β can be a driver of chondrocyte hypertrophy under OA conditions. For example, TGF-β can signal via its alternative detrimental SMAD1/5/9 signaling route. For example, SMAD1/5 can interact with RUNX2 to induce COL10A1 expression, an important hypertrophy gene [6,11,12,13]. In addition, TGF-β can induce many other SMAD-independent signaling routes, such as mitogen-activated protein kinase (MAPK) and phosphoinositide 3-kinase (PI3K) pathways [6]. It is unknown if activation of these SMAD-independent pathways and subsequent activation of corresponding TFs contribute to induction of hypertrophy during osteoarthritis [14,15,16,17,18,19]. Previous research in experimental OA models [20,21,22] and also recent large-scale gene expression data confirm that TGF-β pathways can contribute to OA development and chondrocyte hypertrophy, since expression of *TGFB1*, the gene encoding for TGF-β1, and other TGF-β-related genes is found to be strongly correlated with chondrocytes in pre-hypertrophic and late-OA stages [23,24,25,26].

Importantly, in OA, active TGF-β levels are elevated compared to levels in healthy joints where these are low and only locally load-activated [27,28,29,30,31]. During OA, such elevated TGF-β levels activate different TFs and signaling pathways than in a healthy joint because of the use of different receptors and co-receptors [9,32,33]. Improving our understanding of TGF-β-mediated activation of TFs and signaling pathways in this context will help us to develop new treatment strategies for OA. Therefore, in this study, we examine which transcription factors (TFs) are activated in human chondrocytes, that are stimulated to undergo hypertrophy, by levels of active TGF-β comparable to those found in the synovial fluid in OA joints.

## 2. Materials and Methods

### 2.1. Patient Material

Primary human chondrocytes were isolated from macroscopically intact cartilage obtained from 30 anonymous OA donors (12 males and 18 females, ages 56–84; median 71 years) undergoing total knee arthroplasty (Sint Maartenskliniek, Nijmegen, the Netherlands). Patients were informed, and permission was requested for the anonymized use of this material for research (File CMO 2018-4319). These donors were randomly distributed over the different experiments.

### 2.2. Primary Chondrocyte Cell Culture and Stimulation

To isolate chondrocytes, cartilage slices were digested by incubation with 1.0 mg/mL pronase (Roche Diagnostics, Basel, Switzerland) for 30 min at 37 °C followed by overnight incubation in 1.5 mg/mL collagenase B (Roche Diagnostics) in DMEM/F12 medium (1:1) (Gibco) at 37 °C. The next day, chondrocyte suspension was spun down at 300× *g* for 10 min, washed three times using saline and resuspended in DMEM/F12 (ThermoFisher, Carlsbad, CA, USA) containing 10% fetal calf serum (FCS; Sigma-Aldrich, Saint Louis, Missouri, USA), 100 mg/L sodium pyruvate, 100 U/mL penicillin and 100 µg/mL streptomycin (Gibco, Carlsbad, CA, USA) (complete culture medium). Chondrocytes were plated at a density of 4.7 × 10^4^ cells/cm^2^ in white polystyrene 96-well plates for luciferase reporter assays and in 24-well plates for gene expression experiments and cultured for one week prior to experiments at 37 °C and 5% CO_2_. Medium was refreshed every three days. Before start of experiments, chondrocytes were serum-starved (0% FCS) overnight. Chondrocytes were stimulated with rhTGF-β1 (Peprotech, Cranbury, NJ, USA) for time periods and dosages indicated in figure legends. Inhibitors to block TGF-β-induced signaling were 5 µM SB-505124 (ALK4/5/7 inhibitor, Sigma-Aldrich), 1 µM Tofacitinib (JAK inhibitor, LC laboratories, Woburn, MA, USA) and 10 µM SP-600125 (JNK inhibitor, Calbiochem, San Diego, CA, USA), all dissolved in dimethyl sulfoxide (DMSO) and pre-incubated for 1 h before TGF-β stimulation.

### 2.3. Construction Reporter Plasmids, Virus Production and Transduction

Binding sequences specific to 15 different transcription factors known to activate chondrocyte signaling pathways (Table 1) [8,26,32,33,34] were extended with a spacer and minimal promoter (sequence: *AGAGGGTATATAATGGAAGCTCGACTTCCAG*) (Full sequences in Appendix A) were derived from Promega (Madison, WI, USA) and synthesized by Genecust (Boynes, France). These binding sequences were directionally cloned into the pNL1.2 Nanoluciferase-PEST (NlucP) vector (Promega). NlucP is an unstable luciferase and was chosen because it has very low background signal and is therefore able to detect minimal responses [35]. Subsequently, lentiviral constructs were generated by re-cloning pNL1.2 reporters with the In-Fusing Cloning method (TakaraBio, Kusatsu, Shiga, Japan) in the ClaI restriction site of the pLVX-EF1α-IRES-Puro producer vector (TakaraBio). Viral supernatants were generated in Lenti-X 293T cells (TakaraBio) and the 4th generation lentiviral production system (TakaraBio) using 1 mg/mL polyethylenimine (PEI; Polysciences, Warrington, USA). Viral supernatant was concentrated using Lenti-X concentrator (TakaraBio) according to manufactures’ protocol. Lentiviral concentration was determined using p24 INNOTEST ELISA assay (Fujirebio, Gent, Belgium). Primary OA chondrocytes, cultured for a week to form a monolayer, were transduced for 6–8 h with 500 ng of virus particles per 62,500 cells with 8 µg/mL polybrene in DMEM/F12 medium without FCS, penicillin or streptomycin (protocol was optimized: Appendix A). A selection pressure of 2 µg/mL puromycin (Sigma-Aldrich) for 7 days was used to ensure transgene expression (± 90% efficient transduction using 500 ng of lentivirus per 62,500 cells, results not shown).

### 2.4. Stimulation and Reporter Gene Assays

After lentiviral transduction for 48 h, cells were serum-starved overnight in 0% FCS supplemented DMEM/F12. Transduced and serum-starved cells were stimulated for 6 h with 5 ng/mL TGF-β (with or without 1 h pre-incubation of inhibitors) or an appropriate positive control (Table 1), which is a compound known to activate the construct in a chondrocyte cell line. Cells were lysed 6 h post-stimulation using 30 µl of ultra-pure water. An equal amount of Nano-Glo luciferase reagent (Promega) was added, and luminescence was determined at 470–480 nm at room temperature (CLARIOstar, BMG Labtech, Ortenberg, Germany). Each condition was performed in quadruplicate, and the mean per donor is depicted.

### 2.5. RNA Isolation and Quantitative Real-Time PCR

mRNA was isolated using 500 µL of TRIzol (Sigma-Aldrich), according to manufacturer’s protocol. After isolation, a maximum of 1 µg of mRNA was treated with 1 µL of DNAse (Life Technologies, Carlsbad, CA, USA) for 15 min at room temperature to remove possible genomic DNA, followed by 10 min of inactivation by incubation at 65 °C with 1 µL of 25 mM EDTA (Life Technologies). mRNA was reverse-transcribed to complementary DNA using 1.9 µL of ultrapure water, 2.4 µL of 10× DNAse buffer, 2.0 µL of 0.1M dithiothreitol, 0.8 µL of 25mM dNTP, 0.4 µg of oligo dT primer, 200U M-MLV reverse transcriptase (all Life Technologies) and 0.5 µL of 40 U/mL RNAsin (Promega) and incubated for 5 min at 25 °C, 60 min at 39 °C and 5 min at 95 °C using a thermocycler. Gene expression was measured using SYBR Green Master Mix (Applied Biosystems, Waltham, MA, USA) and 0.25 mM primers (Biolegio, Nijmegen, the Netherlands: see Table 2) with a StepOnePlus real-time PCR system (Applied Biosystems). The amplification protocol was 10 min at 95 °C, followed by 40 cycles of 15 s at 95 °C and 1 min at 60 °C. Melting curves were analyzed to confirm product specificity. To calculate the relative gene expression (−ΔCt), the average of the three reference genes *GAPDH*, *RPL22* and *RPS27A* was used.

### 2.6. Statistical Analysis

Quantitative data of gene expression analysis are expressed as column scatter graphs, and one dot displays the mean value of a technical triplicate sample per donor. Results are represented as mean of multiple donors with corresponding 95% confidence interval (CI). For TF-reporter assays, conditions were investigated in quadruple, and one dot displays the mean of one donor. The graphs show the mean of multiple experiments with corresponding 95% confidence interval (CI). Differences were tested using displayed means with analysis of variance (ANOVA) followed by Bonferroni’s post-test to take multiple comparisons into account. Statistical differences were considered as significant if the *p*-value was below 0.05. All analyses were performed using Graph Pad Prism version 9.0 (GraphPad Software).

## 3. Results

### 3.1. TGF-β Drives a Hypertrophy-like Phenotype in Primary Human OA Chondrocytes

The growth factor TGF-β is important for chondrocyte homeostasis, but can be anti- or pro-hypertrophic, depending on its concentration and on the chondrocyte responsiveness. We established an in vitro model in which TGF-β drives a hypertrophy-like phenotype in primary human OA chondrocytes, which we defined as increased expression of characteristic markers of chondrocyte hypertrophy *RUNX2*, *COL10A1*, *VEGFA* and *IHH* [2], and increased expression of dedifferentiation marker *COL1A1*, which is also related to an OA phenotype [75,76]. For this, we used a concentration of 5 or 10 ng/mL TGF-β, levels comparable to those found in the synovial fluid of OA patients [27,28,29,30,31], and tested whether these concentrations of TGF-β were able to increase expression of these hypertrophy genes. After stimulation of OA chondrocytes of two donors with 5 or 10 ng/mL TGF-β1 for 24 and 48 h, expression of *COL1A1*, *IHH* and *VEGFA* were increased significantly both at 24 and 48 h (Figure 1A). *COL10A1* was only significantly upregulated when stimulated for 48 h with TGF-β1, and *RUNX2* was not significantly upregulated. Because we found no difference between a concentration of 5 or 10 ng/mL TGF-β and the hypertrophy-like phenotype was best induced after 48 h, we continued with 5 ng/mL TGF-β for 48 h for our hypertrophy-like model. We verified the model in OA chondrocytes of another 24 donors, incubated for 48 h with 5 ng/mL TGF-β1 (Figure 1B). mRNA expression of *COL10A1* (+1.147 ∆∆Ct, *p* < 0.001), *RUNX2* (+1.564 ∆∆Ct, *p* < 0.001), *COL1A1* (+2.882 ∆∆Ct, *p* < 0.001), *VEGFA* (+1.189 ∆∆Ct, *p* < 0.001) and *IHH* (+2.592 ∆∆Ct *p* < 0.001) was strongly upregulated, confirming a hypertrophy-like phenotype in OA chondrocytes induced by TGF-β.

### 3.2. TGF-β Activates SMAD3:4, STAT3 and AP1 Signaling in Human OA Chondrocytes

To identify which TFs were activated by TGF-β in OA chondrocytes undergoing hypertrophy, we made use of luciferase-based TF activity assays. Using lentiviral transduction, 15 different TF-based luciferase constructs (Table 1), each of them known to be downstream of chondrocyte signaling pathways [8,26,32,33,34], were transduced into human OA chondrocytes. Importantly, this lentiviral transduction did not affect our TGF-β-induced hypertrophy model in these cells (Appendix A). For every TF-based construct, activation upon TGF-β treatment and a positive control (a compound known to activate the construct in a chondrocyte cell line: Table 1) was assessed in multiple OA chondrocyte donors (Appendix A). Stimulation for 6 h with an appropriate positive control induced SBE-, SIE-, AP1-, NFκB-, CRE-, SRF-, GRE-, SRE-, ISRE- and NFAT5-controlled pathways in human OA chondrocytes. TCF/LEF, PPRE, SOX9, CSL and ARE signaling were not activated by a positive stimulus or were already fully activated in control condition.

TGF-β in a concentration of 5 ng/mL for 6 h did not activate NFκB-, CRE-, SRF-, GRE-, SRE-, ISRE-, NFAT5-, TCF/LEF-, PPRE-, SOX9-, CSL- or ARE-controlled pathways (Appendix A). Importantly, TGF-β did activate SBE, SIE and AP1 signaling (Figure 2 and Appendix A). In all seven donors, 5 ng/mL TGF-β significantly increased SBE-luciferase (Appendix A) activity with an average of a 20.09-fold increase (*p* < 0.001) (Figure 2A). SIE signaling was significantly increased in two out of seven donors (Appendix A), however with a significant average 2.23-fold increase (*p* = 0.0039) (Figure 2B). In six out of seven donors, AP1 luciferase signaling was significantly increased (Appendix A) with a 2.70-fold average increase (*p* = 0.0059) (Figure 2C). These results indicate that these pathways are potentially involved in TGF-β-induced OA pathogenesis and induction of hypertrophy-like differentiation in OA chondrocytes.

### 3.3. TGF-β Activated SMAD3:4, STAT3 and AP1 Signaling Is Blocked by Specific Inhibitors

Specific pathway inhibitors were used to probe the involvement of the SBE, SIE and AP1 pathways in the involvement of the TGF-β-driven hypertrophy-like phenotype. The SMAD binding element (SBE: *GTCTAGAC*) in DNA binds a phosphorylated complex of R-SMAD3 and co-SMAD4, which is formed and transported to the nucleus upon TGF-β receptor type I activation, i.e., ALK4, ALK5 and ALK7 (Figure 3A) [6]. These receptors can be inhibited with the SB-505124 inhibitor, thereby blocking SMAD3 activation [77]. We pre-incubated human chondrocytes transduced with the SBE-luciferase reporter construct for 1 h with 5 µM SB-505124 before stimulation with TGF-β, showing significant inhibition of luciferase induction with both 1 and 5 ng/mL TGF-β (*p* < 0.0001) (Figure 3B).

TGF-β also induced luciferase activation of the SIE-responsive element, which consists of repeats of Interleukin (IL)-6 Sis-Inducible Elements (SIE: *TTCCCGTAA*), also called STAT3 response elements, since it is activated by STAT3 [78]. STAT3 is phosphorylated by receptor-associated Janus kinases (JAKs) in response to many different ligands as IL-6, interferons and fibroblast growth factor (FGF) (Figure 3C). Tofacitinib is an inhibitor of JAK1 and JAK3, thereby interfering with this JAK-STAT pathway [79]. Pre-incubation with 1 µM Tofacitinib completely blocked IL-6-induced SIE signaling (*p* < 0.0001), which we used as a positive control to activate this pathway. In addition, Tofacitinib completely blocked TGF-β-induced SIE signaling (*p* < 0.0001) (Figure 3D).

The other pathway we found to be induced by TGF-β in OA chondrocytes was regulated by the transcription factor Activator Protein 1 (AP1: *TGA G/C TCA*), which is induced in response to a variety of stimuli, including cytokines, growth factors and stress. AP1 is a heterodimer composed of proteins belonging to the c-FOS, c-JUN and activation transcription factor (ATF) protein families, which are activated by mitogen-activated protein kinases (MAPKs) pathways [80,81]. The three separate groups of the MAPKs consist of the ERKs, p38s and the c-JUN N-terminal kinases (JNKs), all activated by different MAPK kinases (MAPKKs) (Figure 3E). We used the JNK inhibitor SP-600125, inhibiting all isoforms JNK1, JNK2 and JNK3, as a possible strategy to interfere with the AP1 pathway. Pre-incubation of 10 µM SP-600125 before stimulation with 20% FCS, which we used as a positive control to activate AP1-targeted signaling, inhibited the AP1-induced luciferase signal for 80,5% (*p* < 0.0001). The TGF-β-induced AP1-luciferase signal was also fully blocked by SP-600125 (*p* < 0.0001) (Figure 3F).

### 3.4. TGF-β-Driven Hypertrophy-like Phenotype in OA Chondrocytes Is Dependent on ALK5 and Is Not Reliant on JAK and JNK

Finally, we wanted to investigate whether blocking these TGF-β-activated pathways can also prevent the TGF-β-driven hypertrophy-like phenotype in human OA chondrocytes, again defined by expression of crucial hypertrophy and dedifferentiation genes. Before stimulation with 5 ng/mL TGF-β, we pre-incubated it with one of the three different pathway inhibitors (and DMSO as the vehicle control) and measured the expression of crucial hypertrophy genes *RUNX2*, *COL10A1* and *IHH* and dedifferentiation gene *COL1A1*. Inhibition of the SBE pathway by blocking ALK4/5/7 kinase activity with SB-505124 abolished the effect of TGF-β on the genes *RUNX2* (−2.782 ∆∆Ct, *p* = 0.0005) (Figure 4A), *COL10A1* (−2.064 ∆∆Ct, *p* = 0.0002) (Figure 4B), *VEGFA* (−2.614 ∆∆Ct, *p* < 0.0001) (Figure 4C) and *COL1A1* (−1.703 ∆∆Ct, *p* = 0.0007) (Figure 4D). This suggests that TGF-β-induced ALK5 kinase activity, which is responsible for SMAD phosphorylation, is involved in the development of a hypertrophy-like phenotype by TGF-β. On the other hand, the JAK-STAT pathway and AP1 signaling via JNK are not involved in induction of TGF-β-driven hypertrophy. Blocking these pathways did not affect the upregulation of hypertrophy and dedifferentiation factors by TGF-β (Figure 4A–D).

## 4. Discussion

Chondrocyte hypertrophy is a hallmark of OA development, and under healthy conditions, tightly regulated by the growth factor TGF-β [6,7,8]. Due to inflammation, cartilage degradation and high levels of proteases, levels of active TGF-β increase in synovial fluid of OA joints, changing the role of this growth factor [27,28]. In this study, we showed that levels of TGF-β similar to those in OA synovial fluid can induce a hypertrophy-like phenotype in human OA chondrocytes, which we defined as increased expression of crucial hypertrophy factors *RUNX2*, *COL10A1*, *VEGFA* and *IHH* and dedifferentiation factor *COL1A1*. We demonstrated that TGF-β activates signaling through the TFs SMAD3:4, STAT3 and AP1 in OA chondrocytes. Although activated, STAT3 and AP1 signaling did not contribute to TGF-β-induced hypertrophic factors, whereas blockage of ALK-5 kinase activity did prevent a hypertrophy-like phenotype induced by TGF-β in OA chondrocytes.

To study the potency of TGF-β in inducing chondrocyte hypertrophy, we made use of primary end-stage OA chondrocytes originating from patients receiving total knee replacement. We stimulated these cells with TGF-β1, which is the predominant form of TGF-β in articular cartilage [7]. Under physiological conditions, TGF-β levels in the joint are predominantly present in a latent form (>98% of total), and TGF-β’s activity requires activation [82]. However, during OA levels of active TGF-β are significantly elevated due to the presence of inflammation in over 50% of all OA patients [83,84]. The activation of macrophages and fibroblasts, cartilage degradation and high levels of proteases in osteoarthritic joints result in increased production and activation of TGF-β [9,28,29]. Unfortunately, most studies have only reported total TGF-β concentrations in synovial fluid without distinguishing between the active and inactive form, which makes it difficult to make a statement about actual active TGF-β levels in situ (overall estimation of 5–10 ng/mL active TGF-β in OA joints) [27,28,29,30,31]. The availability of receptors and co-receptors and differential activation of them with ageing and during OA conditions will also influence sensibility and responsiveness of the chondrocytes to TGF-β [85,86]. These factors combined will contribute to further elevated levels of active TGF-β and different subsequent downstream signaling in OA versus healthy chondrocytes.

In healthy chondrocytes, TGF-β acts as an inhibitor of chondrocyte hypertrophic differentiation and reduces expression of COL10A1 and MMP13 via SMAD3 activation. However, during OA a shift in receptor use and disturbed downstream TGF-β signaling causes a switch of the chondrocyte phenotype [87,88]. Our results illustrate that stimulation of primary human OA chondrocytes with representative levels of TGF-β in the synovial fluid of OA patients causes a hypertrophy-like phenotype in these cells. In this study, we defined a hypertrophy-like phenotype caused by increased expression of the genes *COL10A1*, *RUNX2*, *IHH* and *VEGFA,* which are factors that are associated with calcification of the articular cartilage and characteristic of chondrocyte hypertrophy [2] and increased expression of dedifferentiation marker *COL1A1*, which is also correlated to OA severity [75,76]. Allas et al., who used the same experimental set-up, demonstrated these pro-hypertrophic effects of TGF-β before, also at the protein level [89]. Other late-hypertrophic genes, such as *alkaline phosphatase* (*ALPL*), *MMP13* and *ADAMTS5*, were not upregulated after 48 h of stimulation with TGF-β (Appendix A). The short time period of 48 h of stimulation with TGF-β limited us in the study of expression of these late-hypertrophic catabolic genes and calcification of the extracellular matrix [90,91,92].

*COL10A1* is considered as the standard marker and *RUNX2* as the master transcription factor of chondrocyte hypertrophy [2,34,93,94,95,96,97]. *IHH* is a key regulator of endochondral ossification and is mainly produced and secreted by pre-hypertrophic chondrocytes [98,99,100], while *VEGFA* is mainly found in the late hypertrophic phase, correlating with osteoblast formation [95,101,102,103]. Lastly, TGF-β increased *COL1A1* expression, which is associated with dedifferentiation towards fibrotic chondrocytes. It is present in the superficial layer of the cartilage or near cartilage lesions. Since it is found to be correlated with OA severity and also upregulated by TGF-β levels comparable to those found in OA patients, we included this factor in our study [75,76,104,105,106,107]. Because we saw induction of these genes in our model, we think our model is appropriate for use to study the TGF-β-induced hypertrophy-like phenotype and that the use of this easy and quick in vitro model of the TGF-β-induced OA phenotype could be useful to screen OA treatments.

A number of TFs have been identified to regulate chondrocyte differentiation towards hypertrophy, of which many could also be activated by TGF-β [2,3]. We identified which TFs are activated by TGF-β levels comparable to those in OA patients and whether they have a role in induction of chondrocyte hypertrophy in our in vitro hypertrophy-like model in primary human OA chondrocytes. We used luciferase-based TF activity assays; all of these TFs are known to be downstream of relevant chondrocyte signaling pathways [8,26,32,33,34]. Luciferase assays have been efficiently used to define and quantify the functional connection between a protein (in our case TGF-β) and transcription activation or repression by many TF complexes [35]. Of note, we studied early activation ofTFs by TGF-β stimulation, because we wanted to exclude indirect activation of the TFs, which could occur when TGF-β activates or represses a distinct protein that itself affects transcription. In human OA chondrocytes, we did not observe NFκB-, CRE-, SRF-, GRE-, SRE-, ISRE-, NFAT5- TCF/LEF-, PPRE-, SOX9-, CSL- and ARE-controlled pathways to be activated by prolonged TGF-β stimulation. However, this does not mean that these pathways do not play a role in other OA pathology or even chondrocyte hypertrophy—merely that their direct role in TGF-β-induced hypertrophy is possibly limited. The molecular control of the chondrocyte phenotype is kept in check by dozens of TFs and chondrogenic transcriptional networks, not only by TGF-β [108]. There are various ways that could lead to chondrocyte hypertrophy. For example, SOX9, TCF/LEF (Wnt signaling), NFAT5 and CREB are also found to be associated with chondrocyte differentiation [97,108].

In this study, we found the SIE pathway to be activated by TGF-β in OA chondrocytes. The SIE pathway runs via activation of STAT3 [78]. We reported earlier that TGF-β activates STAT3 signaling itself, and also that TGF-β stimulates IL-6 production by articular chondrocytes, which then also activates STAT3 [109]. In this study, SIE signaling could be completely blocked when pre-stimulated with Tofacitinib, which is an inhibitor that targets JAK kinases upstream of STAT3. Tofacitinib is already clinically effective in rheumatoid arthritis but has not yet been tested in OA patients [110,111,112]. There are indications that JAK-STAT inhibition could be promising to target OA pathology. For instance, Tofacitinib blocked cytokine-induced proteoglycan loss and restored COL2 synthesis in bovine cartilage [113]. Additionally, in experimental OA mouse models, JAK-STAT inhibition appears to have protective effects [65,114,115]. On the other hand, our results demonstrate that Tofacitinib was not able to block TGF-β-induced expression of *RUNX2*, *COL10A1*, *VEGFA* and *COL1A1* in OA chondrocytes, suggesting a limited role for JAK-STAT signaling in TGF-β-driven hypertrophy. It would be interesting to elucidate if targeting JAK-STAT has beneficial roles in joint biology and OA development [116].

Previous research established that the transcription factor AP1 can directly interact with SOX9 and RUNX2, thereby promoting hypertrophic gene expression [117,118]. Therefore, AP-1 signaling was also included in this study, and it was found to be activated by TGF-β stimulation in OA chondrocytes as well. AP1 consists of the subunits c-JUN and c-FOS, which were formed after MAPKs activation. The three separate groups of the MAPKs consist of the ERKs, p38s and the JNKs, which in turn are all activated by different MAPKKS (MEK1-7) [80,81]. Despite major progress in understanding AP1 signaling and function, as well as the identification of small molecules integrating with AP1 signaling pathways, no effective AP1 inhibitors have yet been approved for clinical use [36,119]. In our study, we completely blocked the AP1 pathway by inhibiting JNK with the inhibitor SP-600125, which suggests that AP1 signaling in human OA chondrocytes is dependent on JNK signaling. Nevertheless, we found that blocking JNK did not inhibit the TGF-β-induced hypertrophy-like phenotype in OA chondrocytes. According to the literature, JNK activation is associated with decreased proteoglycan synthesis and enhanced MMP13 expression and is linked to osteoblast proliferation and development of osteosarcoma [39,120,121]. A possible role for AP1 signaling in induction of chondrocyte hypertrophy in OA could still exists, although not in the TGF-β-induced hypertrophy-like phenotype of chondrocytes.

As expected, we found the SBE signaling pathway to be activated by TGF-β in OA chondrocytes, which is regulated by the TFs SMAD3 and SMAD4. Moreover, blocking ALK4/5/7 kinase activity with the inhibitor SB-505124 abolished the effect of TGF-β on hypertrophy-related genes. The type I receptor ALK5 is known to be specific for TGF-β ligands, and ALK4 and ALK7 are thought to mediate signaling via the ligands activin and nodal[114,115,122]. This suggests that receptor SMADs, activated by TGF-β-induced ALK5 phosphorylation, are involved in TGF-β-driven hypertrophy. These receptor SMADs include the homeostatic SMAD2 and SMAD3, as well as SMAD1 and SMAD5. Activation of SMAD3, the transcription factor of the SBE construct, is associated with inhibition of chondrocyte hypertrophy [9,10]. Meanwhile, signaling via SMAD1/5 promotes pro-hypertrophic signaling via RUNX2 and expression of COL10A1 [6,11,12,13]. The detrimental SMAD1/5 signaling route, for which activation was not reflected using our SBE construct, is mostly activated by bone morphogenetic proteins (BMPs) [6]. However, we reported earlier that TGF-β is also able to induce SMAD1/5 signaling, via activation of the ALK5:ALK1 receptor complex, and this activation is also inhibited using SB-505124 [123]. Previous studies have shown that this TGF-β-induced SMAD1/5 signaling does not act via BMP response elements (BREs), since TGF-β failed to induce BRE-luciferase activity [124,125], and therefore we have not measured these SMADs with a BRE-TF construct. The importance of ALK5 kinase activity in SMAD1/5 signaling was already demonstrated by Goumans et al. [120] Of note, it is known that high levels of TGF-β during OA conditions skew more to ALK1/SMAD1/5 signaling in chondrocytes, and ageing a shift towards ALK1 is observed, leading to increased MMP13 expression [85]. Additionally, Qu et al. demonstrated that high doses of TGF-β contribute to degeneration of nucleus pulposus cells in the intervertebral disc via upregulation of ALK1 and SMAD1/5 [121]. Moreover, gene expression analysis in experimental OA models and human OA samples showed upregulation of the TGF-β co-receptor Cripto, which participates in a TGF-β-ALK1-Cripto receptor complex, thereby inducing SMAD1/5 signaling in chondrocytes and induction of hypertrophy [126]. Thus, the impact of increased TGF-β activation in articular cartilage may move from homeostatic SMAD2/3 signaling towards pathological SMAD1/5 signaling, depending on its levels, context and receptor usage.

In conclusion, the TGF-β-induced hypertrophy-like phenotype in human OA chondrocytes is dependent on ALK5 kinase activity, of which the SMADs are a prominent target. Although STAT3 and AP1 signaling are also activated by levels of TGF-β similar to those present in OA patients, these appear not to contribute to TGF-β-induced expression of hypertrophic and dedifferentiation genes. Future studies should indicate whether this increased expression of OA genes due to high levels of TGF-β is a result of increased SMAD1/5 signaling activated by ALK5 kinase. TGF-β is of great importance for chondrocyte homeostasis in healthy cartilage but at the same time also a driver for hypertrophy and dedifferentiation in OA circumstances. Therefore, improvement of our understanding of the different subsets of TGF-β-mediated signaling and how they function differently in healthy versus OA chondrocytes is crucial. Inhibition of chondrocyte hypertrophy-like changes by use of inhibitors targeting specific hypertrophy-inducing pathways in TGF-β-signaling might be a promising therapeutic target for slowing down OA progression.

## Figures and Tables

**Figure 1 cells-11-01232-f001:**
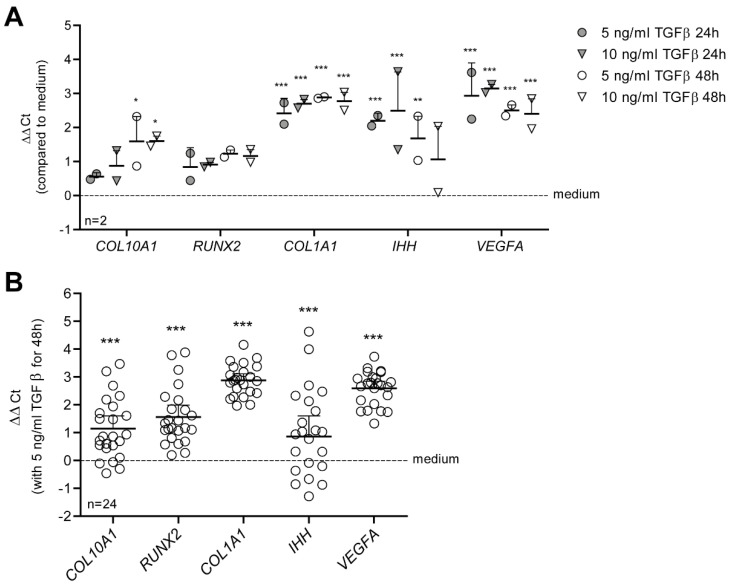
TGF-β1 drives a hypertrophy-like phenotype in primary human OA chondrocytes. Elevated levels of TGF-β that are comparable to those in synovial fluid of OA patients increase expression of crucial hypertrophy genes. (**A**) Human chondrocytes of two donors were cultured in monolayer and stimulated with 5 or 10 ng/mL TGF-β1 for 24 or 48 h. This resulted in induction of hypertrophy-like differentiation, as measured by increased expression of relative collagen type 10 (*COL10A1*), runt-related transcription factor 2 (*RUNX2*), alpha-1 collagen type 1 (*COL1A1*), Indian hedgehog (*IHH*) and vascular endothelial growth factor A (*VEGFA*) using qPCR. (**B**) These results were verified in chondrocytes of another 24 donors, stimulated for 48 h with 5 ng/mL TGF-β1. Data are plotted as mean ± 95% CI with each dot representing the average of three technical replicates in one chondrocyte donor. Statistical analysis was performed using a repeated measures two-way (**A**) or one-way (**B**) ANOVA with Bonferroni’s post hoc test: * *p* ≤ 0.05; ** *p* ≤ 0.01; *** *p* < 0.001.

**Figure 2 cells-11-01232-f002:**
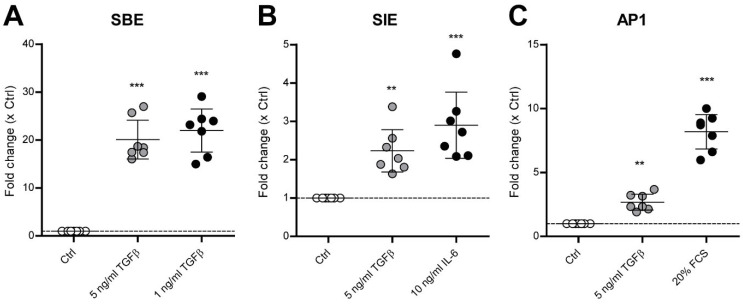
TGF-β1 activates SMAD3:4, STAT3 and AP1 signaling in human OA chondrocytes. Primary human OA chondrocytes, cultured for a week to form a monolayer, were transduced for 6–8 h with 500 ng of lentivirus per 62,500 cells. After 48 h, the transduced cells were serum-starved overnight and then stimulated with a positive control (black dots) or with 5 ng/mL TGF-β1 (gray dots) for 6 h. Luminescence was determined at 470–480 nm, and fold change compared to medium-stimulated cells (Ctrl, white dots) was calculated. TGF-β1 (gray dots) activated cell signaling regulated by the transcription factors (**A**)SMAD3:4 (SBE: SMAD binding element), (**B**) STAT3 (SIE: Interleukin(IL)-6 sis-inducible element or STAT3 response element) and (**C**) AP1 (Activation Protein 1). Data are plotted as mean ± 95% CI, with each dot representing the average of a quadruple sample in one chondrocyte donor (total *n* = 7). Additionally, the induction of the pathways with positive controls is reported (black dots), which were 1 ng/mL TGF-β1 for the SBE pathway, 10 ng/mL IL-6 for the SIE pathway and 20% fetal calf serum (FCS) for the AP1 pathway. Statistical analysis was performed using one-way ANOVA with Bonferroni’s post hoc test: ** *p* ≤ 0.01; *** *p* < 0.001.

**Figure 3 cells-11-01232-f003:**
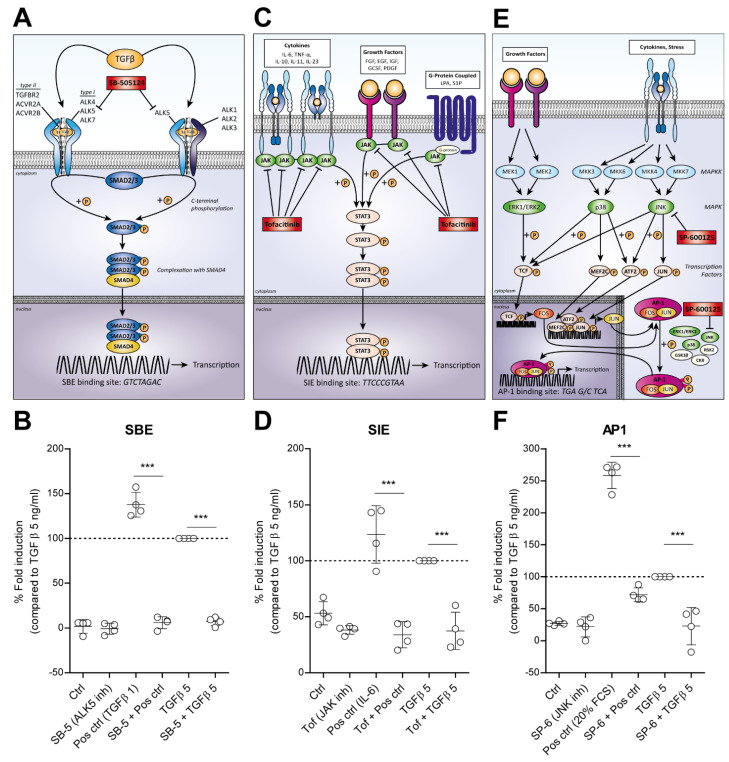
TGF-β1-stimulated SMAD3:4, STAT3 and AP1 signaling are blocked with pathway inhibitors. Specific pathway inhibitors were used to target the SBE, SIE and AP1 pathways. (**A**) The SBE pathway starts with TGF-β ligand binding, which induces formation of a receptor complex containing type I (ALK4/5/7) and type II receptors. Receptor-SMAD3 is recruited and activated by C-terminal phosphorylation (P). A complex is formed with the common SMAD4 and this translocates to the nucleus where it binds the DNA to the SMAD3:4 binding element (SBE: *GTCTAGAC*), activating transcription and luciferase signal. SB-505124 (SB-5) is an ALK-4/5/7 inhibitor interfering with SMAD-dependent signaling. (**B**) Pre-incubation of human chondrocytes (*n* = 4) transduced with the SBE-luciferase reporter construct for 1 h with 5 µM SB-5 before stimulation with TGF-β1 for 6 h, results in significant inhibition of luciferase induction with both 1 (Pos ctrl) and 5 ng/mL TGF-β. (**C**) The SIE pathway consists of repeats of IL-6 sis-inducible elements (SIE: *TTCCCGTA*) and is also called STAT3 response element since it is activated by STAT3. STAT3 is phosphorylated (P) by receptor-associated Janus kinases (JAK) in response to many different ligands. Tofacitinib (Tof) is an inhibitor of JAK1 and JAK3, thereby interfering with this JAK-STAT pathway. (**D**) Pre-incubation of human chondrocytes (*n* = 4), transduced with the SIE-luciferase reporter construct, with 1 µM of completely blocked IL-6 (Pos ctrl) and TGF-β1 (5 ng/mL) induced SIE signaling. (**E**) The AP1 pathway is regulated by transcription factor Activator Protein 1 (AP1: *TGA G/C TCA*), which is a heterodimer composed of FOS, JUN and ATF protein families. These are activated by MAPKs pathways, consisting of the ERKs, p38s and JNKs. SP-600125 (SP-6) is an inhibitor of all JNK isoforms, thereby interfering with this AP1 pathway. (**F**) Pre-incubation of primary human OA chondrocytes, transduced with the AP1-luciferase reporter construct with 10 µM SP-6 blocked luciferase induction with 20% FCS (Pos ctrl) or 5 ng/mL TGF-β1. Data are plotted as mean ± 95% CI with each dot representing the average of four technical replicates in one chondrocyte donor; in total, chondrocytes from four different donors were used, as seen in (**B**,**D**,**F**). Luciferase activity was measured relative to experimental condition stimulated with 5 ng/mL TGF-β, as set at 100%. Statistical analysis was performed using one-way ANOVA with Bonferroni’s post hoc test: *** *p* < 0.001. ALK = ALK tyrosine kinase receptor, ACVR2 = Activin type-2 receptor, GF = growth factor, IL = Interleukin, JAK = janus kinase, JNK = c-JUN N-terminal kinase, MAPK = mitogen-activated protein kinase, STAT3 = Signal Transducer and Activator of Transcription 3, TGFBR = TGF-β receptor.

**Figure 4 cells-11-01232-f004:**
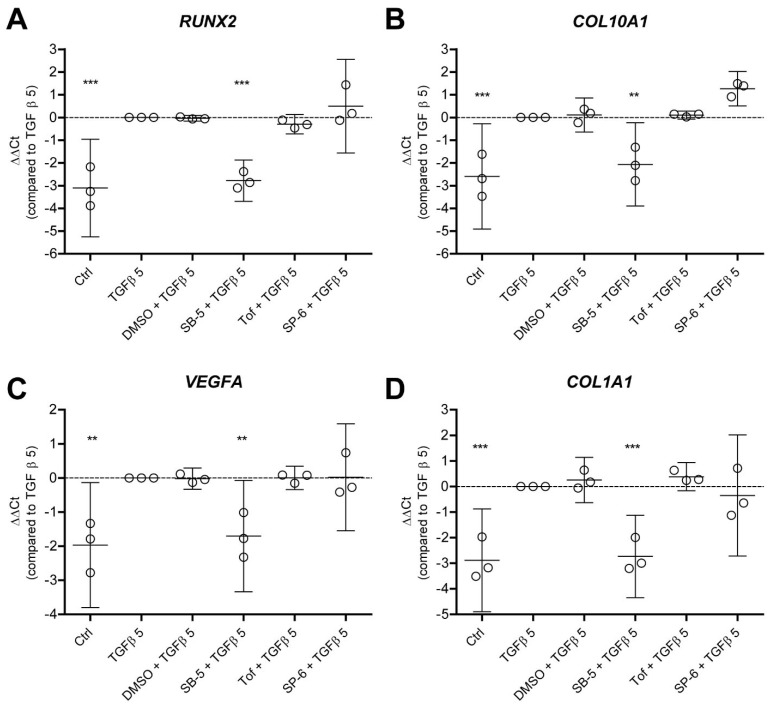
A hypertrophy-like phenotype in OA chondrocytes, induced by TGF-β, is dependent on ALK5 and is not reliant on JAK and JNK. Elevated levels of TGF-β that are comparable to those in synovial fluid of OA patients induce expression of hypertrophic factors in human OA chondrocytes of three different donors. This is defined by increased mRNA expression of (**A**) *runt-related transcription factor 2* (RUNX2), (**B**) *relative collagen type 10* (*COL10A1*), (**C**) *vascular endothelial growth factor A* (*VEGFA*) and (**D**) *alpha-1 collagen type 1* (*COL1A1*) after stimulation with 5 ng/mL TGF-β1 for 48 h. Specific pathway inhibitors were used to target the SBE, SIE and AP1 pathways, which were also activated by these levels of TGF-β. Cells were pre-incubated with 5 µM SB-505124 (SB-5: ALK5 kinase activity inhibitor), 1 µM Tofacitinib (Tof: JAK inhibitor) or 10 µM SP-600125 (SP-6: JNK inhibitor) before stimulation with 5 ng/mL TGF-β1. Only blocking ALK5 kinase activity with SB-5 prevented the effect of TGF-β on hypertrophy genes, whereas inhibition of JAK and JNK did not. Data are plotted as mean ± 95% CI with each dot representing the average of three technical replicates in one chondrocyte donor (total of 3 different chondrocyte donors were used). Statistical analysis was performed using repeated measures one-way ANOVA with Bonferroni’s post hoc test: ** *p* ≤ 0.01; *** *p* < 0.001.

**Table 1 cells-11-01232-t001:** Transcription factor (TF) elements used for luciferase TF activity constructs, each of them known to activate relevant chondrocyte signaling pathways.

Pathway	TF(s)	TF Element (Abbreviation Construct)	Positive Control	References
MAPK/JNK	c-FOS:c-JUN	Activator Protein 1 response element (AP1)	FCS (20%)	[36,37,38,39]
SOX9	SOX9	SRY-box transcription factor 9 response element (SOX9)	Forskolin (10 µM)	[40,41,42]
NFκB	NFκB:p65	Nuclear factor κ B response element (NFκB)	IL-1β (1 ng/mL)	[43,44,45]
Notch	CSL	CBF1/RBPJκ/Suppressor of Hairless/Lag-1 response element (CSL)	FCS (20%)	[46,47,48,49]
cAMP/PKA	CREB	Cyclic AMP response element (CRE)	Forskolin (10 µM)	[41,50,51]
Glucocorticoid signaling	Glucocorticoid receptor	Glucocorticoid receptor response element (GRE)	Dexamethasone (10 µM)	[50,52,53,54]
TGF-β	SMAD3:SMAD4	SMAD binding element (SBE)	TGF-β1 (1 ng/mL)	[8,9,55]
Wnt	TCF/LEF	T-cell factor/lymphoid enhancer factor family response element (TCF/LEF)	Wnt3a (200 ng/mL)	[51,56,57]
INF-α	STAT1:STAT2	Interferon-stimulated response element (ISRE)	IFN-α (100 ng/mL)	[58,59]
IL-6	STAT3:STAT3	Sis-inducible element (SIE)	IL-6 (10 ng/mL)	[55,60,61,62]
MAPK/ERK	ELK-1:SRF	Serum response element (SRE)	FCS (20%)	[63,64]
RhoA	SRF	Serum response factor (SRF)	FCS (20%)	[63,65]
Oxidative stress	NRF2	Antioxidant response element (ARE)	H_2_O_2_ (50 µM)	[66,67,68]
Calcium/calcineurin: hyperosmotic signaling	NFAT5	Nuclear factor of activated T-cells 5 response element (NFAT5)	+100 milliosmole	[69,70,71]
PPARγ	PPARγ	Peroxisome Proliferator activated receptor-γ response element (PPRE)	Rosiglitazone (20 µM)	[72,73,74]

**Table 2 cells-11-01232-t002:** Primer sequences as used in this study.

Gene	Forward Sequence	Reverse Sequence
*GAPDH*	*ATCTTCTTTTGCGTCGCCAG*	*TTCCCCATGGTGTCTGAGC*
*RPL22*	*TCGCTCACCTCCCTTTCTAA*	*TCACGGTGATCTTGCTCTTG*
*RPS27A*	*TGGCTGTCCTGAAATATTATAAGGT*	*CCCCAGCACCACATTCATCA*
*RUNX2*	*GCAAGGTTCAACGATCTGAGA*	*TTCCCGAGGTCCATCTACTG*
*COL10A1*	*TTTTACGCTGAACGATACCAAATG*	*CTGTGTCTTGGTGTTGGGTAGTG*
*COL1A1*	*AGATCGAGAACATCCGGAG*	*AGTACTCTCCACTCTTCCAG*
*VEGFA*	*CAGGGAAGAGGAGGAGATGAGA*	*GCTGGGTTTGTCGGTGTTC*
*IHH*	*CAATTACAATCCAGACATCATCTTCA*	*CGAGATAGCCAGCGAGTTCAG*

## Data Availability

Not applicable.

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
