# Peer review of "Identification of Transcription Factors Responsible for a Transforming Growth Factor-β-Driven Hypertrophy-like Phenotype in Human Osteoarthritic Chondrocytes"

_cells, 2022, doi:10.3390/cells11071232_

Round 1

Reviewer 1 Report

The manuscript entitled "Identification of transcription factors responsible for a transforming growth factor-β driven hypertrophy-like phenotype in human osteoarthritic chondrocytes" by Thielen et al. elucidates which transcription factors are involved in TGF-β dependent induction of chondrocyte hypertrophy in OA, which could be important for future therapeutic intervention.

By extensive cell culture studies with primary chondrocytes from OA patients and reporter gene assays ALK-5 induced SMAD signalling was detected to be essential. 

The study is original and provides readers with new aspects of TGF-β signaling in OA. However, some points remain unclear and require revision. For example, the study addresses the induction of the "hypertrophy-like chondrocyte phenotype" but includes the expression analysis of COL1A1 in all figures. The general accepted opinion is, that COL1A1 is a marker for dedifferentiation to a fibroblast-like phenotype and is unrelated to hypertrophy. 

In addition, the entire study is based only on gene expression analyses. A selective check at the protein level would support the statements made more strongly. 

Reviewer 2 Report

Cells-1634175

Comments to the Author
This research attempts to identify signaling pathways responsible for a transforming growth factor-β1 (TGF-β1), which drive hypertrophy-like phenotype in human osteoarthritic (OA) chondrocytes. The authors confirmed that TGF-β1 increased gene expression of RUNX2, COL10, COL1A1, VEGF, and IHH. Using luciferase assays, they observed that the expression of these genes positively correlated to SMAD3/4, STAT3, and AP1 signaling pathways. Understanding the function and signaling pathways of TGF-β1 in chondrocytes is of considerable interest in understanding the pathogenesis of OA. However, this study is insufficient in elucidating a novel function of TGF-β1 in OA and has shortcomings, which will be outlined below:

  1. No data regarding chondrocyte differentiation is provided to support the function of TGF-β1 in OA chondrocytes, although they focused on “hypertrophy-like phenotype.” Did the authors examine the alcian blue staining and/or calcification in OA chondrocytes after TGF-β1 stimulation?
  2. If the authors focus on hypertrophic chondrocytes, did they examine the expression of other hypertrophic marker genes, for example, NOTCH1, NOTCH2 MMP13, OSX, FGFR3, or WNT5B? The authors claimed that RUNX2, COL10, COL1A1, VEGF, and IHH are characteristic markers for chondrocyte hypertrophy or crucial hypertrophy factors (lines 16, 158, 327), citing reference [2], but COL1A1 is not a marker of chondrocyte hypertrophy. The reference [2] did not mention
  3. Please provide details of the artificial binding sequences specific for 15 TFs (Table 1). Please specify tandem repeats of binding sequences. This reviewer recommends inserting references to give more power and reliability to the information.

Minor comments:

  1. The wording is not sufficiently precise in places. Especially, the authors should carefully check and use only HUGO-approved gene IDs (example, (italics) TGFB1). COL10 and VEGF are not the gene ID (name). Please correct.
  2. I recommend inserting references in the sentence (line 156).
  3. I recommend inserting references in the sentence (line 178). Ref [35] does not refer to all 15 TFs in the text.

Reviewer 3 Report

In this manuscript, the authors attempted to identify transcription factors responsible for TGFbeta dependent chondrocyte hypertrophy. They used human OA chondrocytes isolated from OA patients and found that SMAD, STAT3, and AP-1 pathways were involved in hypertrophic gene expression induced by TGFbeta.

The study was carefully performed and the data presented are convincing. However, there are some important issues that should be addressed.

  1.  Mef2c is a critical transcription factor for chondrocyte hypertrophy (Dev Cell. 2007 Mar;12(3):377-89). The authors should examine whether TGFbeta induces MEF2C expression.
  2. To strengthen the manuscript, the authors are encouraged to examine catabolic gene expression including MMP13 and ADAMTS which contribute to OA development and progression.
  3. Recently, Boer et al identified ‘High-confidence effector genes for OA by conducting GWAS meta-analysis across 826,690 individuals (Cell 184(18):4784-4818, 2021). The authors cite this article and discuss the pathological association between TGFbeta and OA genes.

Round 2

Reviewer 2 Report

N/A.

Author Response

Dear reviewer,

Thank you for your time to review our manuscript and that you seem to be satisfied with our answers and the manuscript.

Kind regards,

the authors